# Analysis of the Results of Pulsed Processing of Melts

**Valerii Krymsky [1],\*, Nataliya Shaburova [2]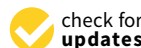 and Ekaterina Litvinova [1]**

[1] Department "Theoretical Foundations of Electrical Engineering", South Ural State University, Lenin prospect 76, Chelyabinsk 454080, Russia; Litvinovaev@susu.ru

[2] Department of Materials Science and Physical Chemistry of Materials, South Ural State University, Lenin prospect 76, Chelyabinsk 454080, Russia; shaburovana@susu.ru

\* Correspondence: krymskiivv@susu.ru; Tel.: +7-351-267-90-13

**Abstract:** The paper presents the results of a relatively new method of external action on melts developed by the authors. The essence of the technology is the impact on melts before casting with electromagnetic pulses (EMP) of short duration (1 ns) and with high pulse power (2 kW). To create electromagnetic pulses, a generator with the following characteristics was used: pulse amplitude of 10 kV, the leading edge of the pulse was 0.1 ns, pulse duration of 1 ns, pulse repetition rate at 1 kHz, and a calculated pulse power of 2 MW. A distinctive feature of the generator used was its low power consumption of 50 Watts. The results of processing low-melting melts of the Al–50Pb, Bi–38Pb, and Bi–18Sn–32Pb systems presented in the work indicated that EMP treatment led to the occurrence of equilibrium crystallization of the metal, increasing its density. In addition to the experimental results, a theory is provided to explain the mechanism of the influence of pulse processing on the properties of metals of these and other systems previously studied by the authors.

**Keywords:** external physical effects on metal melts; nanosecond electromagnetic pulses; ultrasonic treatment of metals; fusible alloys; mechanism of the influence of pulse processing

---

## 1. Introduction

Various methods are known for using external physical fields for processing molten metals and alloys. Traditionally, ultrasound, electromagnetic field treatment, mechanical stirring, and vibration are used. Despite the differences in the nature of these methods, the effects they exert on the properties of metals turn out to be similar. The microstructure is refined, the mechanical characteristics of the metal are improved, and degassing and uniform crystallization occur. In the case of ultrasound treatment, these changes occur due to vibration and cavitation [1–4]. With electromagnetic field treatment, this is due to the refinement of clusters by vortex melt flows [5–8]. In the case of vibration and mechanical stirring, this is due to the collision and refinement of the crystallization nuclei [9,10]. Each of these methods has its own characteristics and disadvantages. A common disadvantage for all of them is the limited volume of the processed melt.

Our proposed method of processing metal melts, electromagnetic pulse (EMP) processing, is universal and has been tested both on non-ferrous pure metals and alloys, as well as on steels and cast irons. As experiments show, the mass of the processed metal can vary from several hundred grams to several tons. We have presented the results of EMP processing of pure aluminum and zinc in a previous publication [11]. Studies have shown that EMP processing decreases porosity and electrical resistivity while increasing density and hardness of technically pure aluminum. Interesting results are also observed for zinc. In our experiments, ball anodes were cast using standard technology and using EMP processing. We noted a change in the nature of crystallization of castings and an increase in their density due to the almost complete absence of shrinkage defects in the EMP-treated metal.

A previous publication [12] describes the results of processing 3 kg of the Al–6Si silumin alloy and 300 kg of Cu–9.4Al–3.4Fe bronze. For all alloys, we noted the positive effect of EMP processing on mechanical properties (a simultaneous increase in strength and ductility). Additionally, we found an increase in the solubility of the main alloying component in the alpha phase of the matrix, but the phase composition of the alloy was preserved. The study in [13] shows the results of processing 1.9 tons of 1035 steel. Applying EMP processing to molten steel leads to an increase in strength, ductility, and toughness. In addition, we observed 30% refinement of the primary austenitic grain.

It should be noted that only two laboratories in the world are studying the effect of electromagnetic pulses on the properties of melts and crystallized metals. This can be explained by the uniqueness of the generator used to create electromagnetic pulses. Traditionally, generators of this kind have been used to create radar systems and were never used to alter materials.

In previous papers, the effect of EMP on melts with a sufficiently high melting point (700–1500 °C) has been studied. For this paper, it was of interest to determine the effects of EMP treatment on melts with a lower melting point. Therefore, for the present study, we chose alloys with a low melting point and that had a different nature of component interaction. Alloys of the Al–Pb system crystallize by monotectic reaction and essentially do not form solid solutions, and alloys of the Bi-Pb and Bi–Sn–Pb systems crystallize by eutectic reaction and form solid solutions. Because we have accumulated extensive experimental material from past studies, theoretical explanation of the observed phenomena is necessary. In this regard, the discussion section presents our developed theory and the mechanism of influence of EMP processing on the properties of metals.

## 2. Materials and Methods

Alloys were obtained by fusing pure components. First, a sample of the most refractory component was melted in the furnace, then lighter components were added. The composition of the metal being processed, determined by the ladle sample before casting, is shown in Table 1. The experimental conditions are also shown.

**Table 1.** Experimental conditions.

| Alloy Composition, wt % | Alloy Mass, kg | Treatment Duration, min | EMP Processing Temperature, °C |
|---|---|---|---|
| Al–50% Pb | 0.25 | 15 | 730 |
| Bi–38% Pb | 1.5 | 10 | 300 |
| Bi–18% Sn–32% Pb | 1.5 | 10 | 300 |

To create electromagnetic pulses, a generator with the following characteristics was used: pulse amplitude was 10 kV, the leading edge of the pulse was 0.1 ns, pulse duration was 1 ns, pulse repetition rate was 1 kHz, and calculated pulse power was 2 MW. A distinctive feature of the generator used is its low power consumption of 50 Watts. The appearance of the generator and the installation diagram for processing the melts are shown in Figure 1.

Two processes for each alloy were carried out, during which the heating temperature and melt holding time were identical. In one case, the metal was melted in a resistance furnace to a predetermined temperature, mixed, and held for 20 min to equalize the temperature. Next, the melt was EMP-treated for 10–15 min. The duration of the EMP treatment on melts was chosen according to the results of numerous experimental studies [14]. Further, the metal was poured into copper molds. For all alloys, graphite crucibles with a height of 150 mm and a diameter of 80 mm were used. In the second case, the procedure was of the same duration, but during the exposure the melt was not processed. The melt temperature was controlled by a chromel-alumel thermocouple. The copper molds were cooled at room temperature. Because both alloys (processed and not processed by EMP) were cooled under identical conditions, we did not record the temperature during cooling.

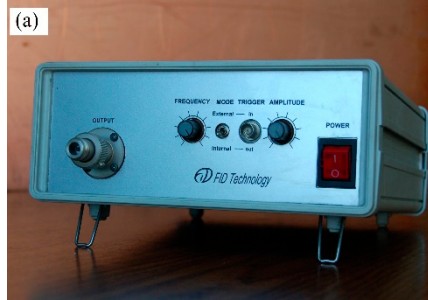 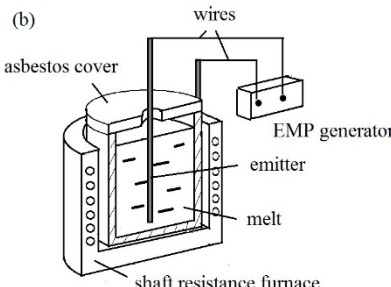

**Figure 1.** Electromagnetic pulse generator (**a**) and installation scheme for processing melts (**b**). EMP: electromagnetic pulse.

This paper presents the averaged results for three experiments for each group of alloys. The scatter of the obtained values in the experiments was not significant.

Sections of the samples were studied using an Axio Observer D1.m optical inverted metallographic microscope (Carl Zeiss Microscopy GmbH, Jena, Germany) equipped with the hardware-software complex for image analysis Thixomet Pro (Thixomet Pro, Thixomet Company, Saint Petersburg, Russia).

Analysis of the chemical composition of the structural components of the samples was carried out on a JEOL JSM-6460LV scanning electron microscope (JEOL, Tokyo, Japan) equipped with an energy dispersive spectrometer (Oxford Instruments, Abingdon, United Kingdom) for qualitative and quantitative X-ray microanalysis (MRSA).

The density of the samples was measured through hydrostatic weighing on an HR-AZG analytical scale (A&D Co. Ltd., Tokyo, Japan) with a measurement accuracy of 0.1 mg.

## 3. Results

### 3.1. Al–50Pb Alloy Processing Results

Figure 2 shows the templates of the cast ingots.

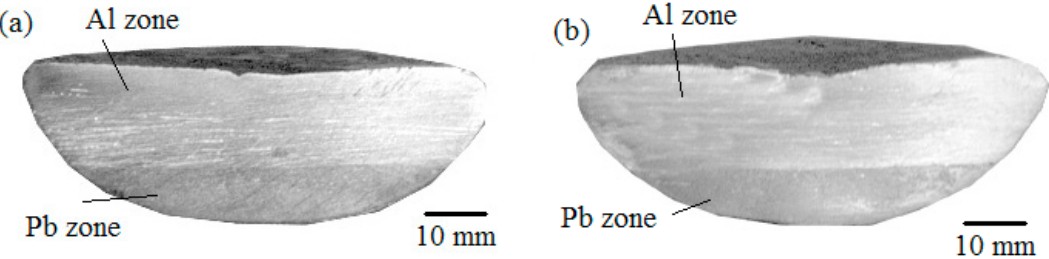

**Figure 2.** Ingots of Al–50%Pb alloys: (**a**) without treatment; (**b**) ingot after EMP processing.

Because Al–50Pb alloy refers to a system that does not form solid solutions, during casting, in the resulting ingot lead (more dense) was deposited under the aluminum (lower density), forming two clearly distinguishable layers in both ingots, as shown in Figure 2. The zone of contact between these two layers was of particular interest for our study.

Panoramas of the microstructures of the untreated and treated alloy are shown in Figure 3; the lead and aluminum layers of the ingot were structurally heterogeneous. Globules of the lead phase were present in the aluminum layer and vice versa (light phase in the lead layer of the ingot). Estimation of the number of structural components was carried out by means of the hardware-software complex of image analysis Thixomet Pro. The measured proportion of the lead phase in the aluminum layer of the ingot (upper layer) in the EMP-treated metal decreased from 5% to 1–2%, compared with the untreated EMP metal. In addition, the particle size of this phase decreased from 10–15 microns to 3–7

microns. Similar changes were observed for inclusions of the aluminum phase in the lead (lower) layer of the ingot.

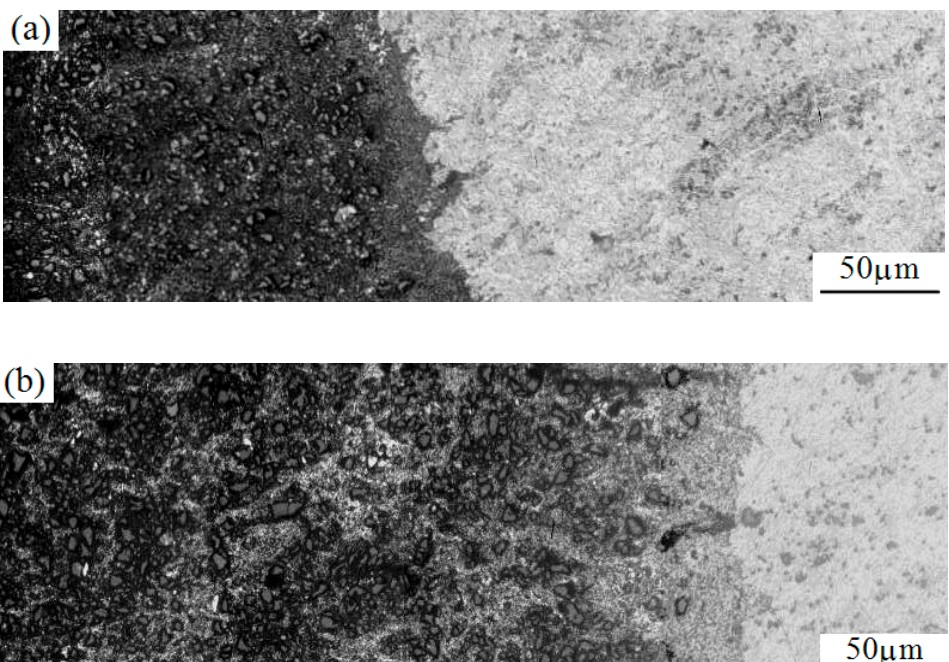

**Figure 3.** Microstructure of the transition zone of alloy ingots of the Al–50Pb system in (**a**) the initial state and (**b**) after EMP treatment. The light field is the aluminum phase, and the dark field is the lead phase.

To determine the width of the zone of interpenetration of the alloy components on a scanning electron microscope (SEM), we studied the distribution of elements by scanning along a line crossing the fusion zone. Figure 4 shows the results of a chemical analysis of the fusion zone of the two ingots. The width of the zone of variation in the radiation intensities of the K-$\alpha_1$ line of aluminum and M-$\alpha_1$ of lead (see Figure 4a,b) led us to conclude that there was a transfer of elements across the contact boundary in both directions.

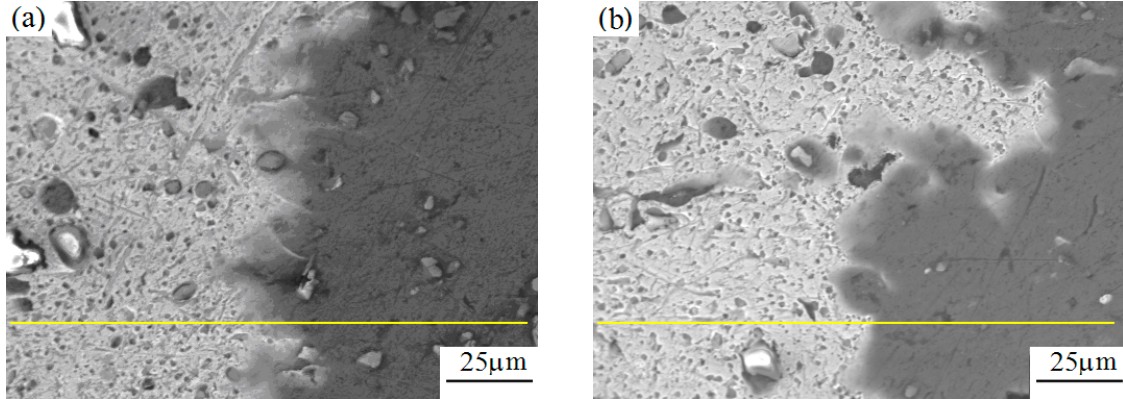

**Figure 4.** *Cont.*

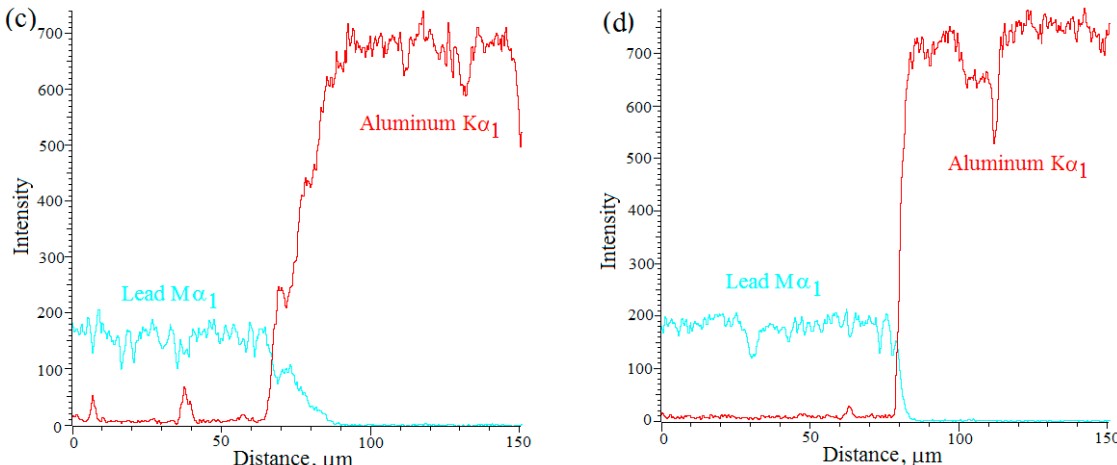

**Figure 4.** The analysis section and the distribution of chemical elements at the fusion boundary of the ingot (**a,c**) without treatment, and (**b,d**) after EMP processing.

Studies have shown that the total width of the fusion zone decreased from 24.1 to 7.5 µm (for metal without treatment and with EMP processing, respectively). In this case, the depth of the zone of allocation of excess phases of lead into the aluminum matrix decreased from 21.6 to 6.25 µm, and aluminum into the lead matrix from 2.5 to 1.25 µm.

We determined the intensities of Al and Pb elements with a scanning electron microscope. The percentage of elements in the fusion zone can be quantified by the reflection intensity of the corresponding lines by taking the intensity of its lines in the aluminum layer as the level of lead at zero, and the intensity of its lines in the lead zone as 100%. The resulting distribution of lead concentration in the aluminum layer of the ingot in the initial and EMP-processed alloys is shown in Figure 5.

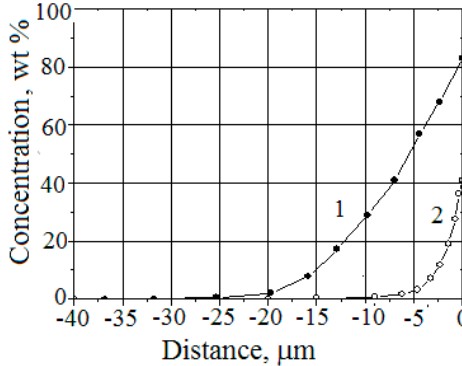

**Figure 5.** The lead content in the aluminum layer of the ingot (1) without treatment and (2) after EMP processing (the zero coordinate is the visible boundary between the lead and aluminum).

From the obtained concentration dependence, we see that the depth of lead concentration in the interpenetration zone changed more intensively for EMP-treated metal. In addition, after EMP treatment, the concentration of lead at the visible boundary of the contact decreased markedly from 82 to 41 wt %.

### 3.2. Bi–38Pb Alloy Processing Results

Typical images of the microstructure of alloys obtained in the study on SEM are shown in Figure 6.

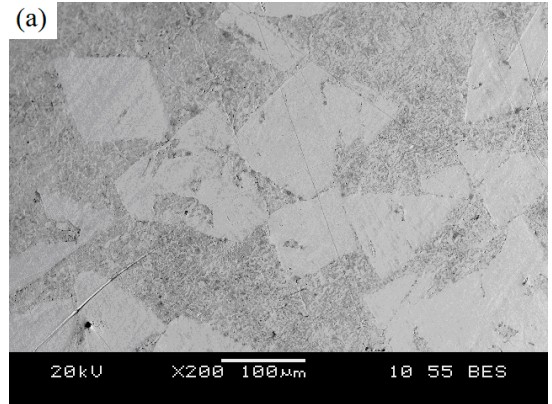 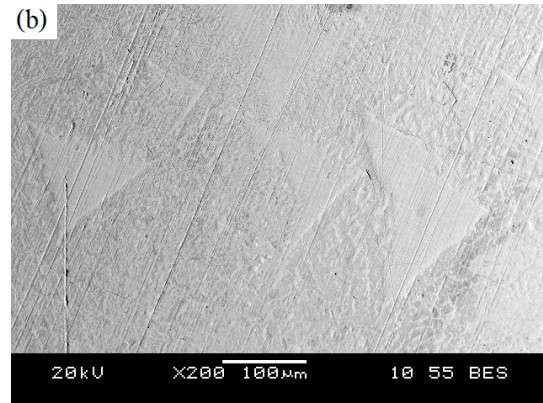

**Figure 6.** Microstructure of the Bi-38Pb sample (**a**) without EMP treatment and (**b**) after EMP processing.

Studies of the microstructure of a sample of a two-component untreated alloy showed that the amount of eutectic in the sample structure did not exceed 45–50%, and primary bismuth crystals did not exceed 50–55% (see Figure 6a).

According to the equilibrium phase diagram, Bi–Pb is a two-component alloy that is not treated by EMP containing an average of 62 wt %. Bi is hypereutectic. The phase composition of hypereutectic alloys in the Pb–Bi system was represented by two phases: primary bismuth crystals crystallizing from the liquid phase and a double lead-bismuth eutectic crystallizing at 125 °C. The amount of lead-bismuth eutectic in the structure of an alloy of the specified composition at equilibrium crystallization should be about 87%, and 13% for primary bismuth crystals.

The noted phase ratio indicated the occurrence of nonequilibrium crystallization, which was most likely due to accelerated cooling.

The two-component alloy processed by EMP had a noticeably smaller number of primary bismuth crystals in the structure—no more than 10% (see Figure 6b). Therefore, for an alloy treated with EMP, we can speak about the occurrence of equilibrium crystallization. At the same time, it should be noted that the size of primary bismuth crystals in the treated metal changed little in comparison with the untreated metal.

Processing melts by EMP led to a decrease in the porosity of the metal, which manifested itself in an increase in its density. The density measurement results showed that EMP treatment increased the density of the alloy; for the raw metal, the density was 10.42 g/cm$^3$, for the treated metal, it was 10.54 g/cm$^3$.

### 3.3. Bi–18Sn–32Pb Alloy Processing Results

Figure 7 shows the characteristic microstructures of samples of a ternary alloy, untreated and treated with EMP. The initial and processed EMP samples had three structural components—primary bismuth crystals bordered by a double tin-bismuth eutectic and a triple lead-tin-bismuth eutectic located on the periphery.

The quantitative ratio of the phases varied significantly—in the structure of the EMP-treated alloy, there were no primary bismuth crystals, and the proportion of double eutectic decreased from 60% to 40% (see Figure 7b).

As in a two-component alloy, a decrease in porosity and an increase in density was observed in a three-component alloy treated with EMP. The measured density of the Bi–18Sn–32Pb alloy treated with EMP was 9.91 g/cm$^3$, whereas for untreated it was 9.78 g/cm$^3$.

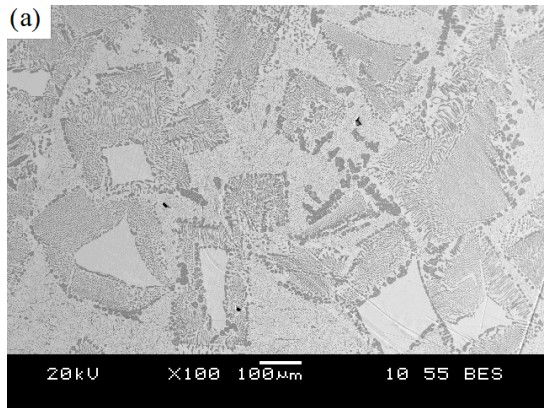
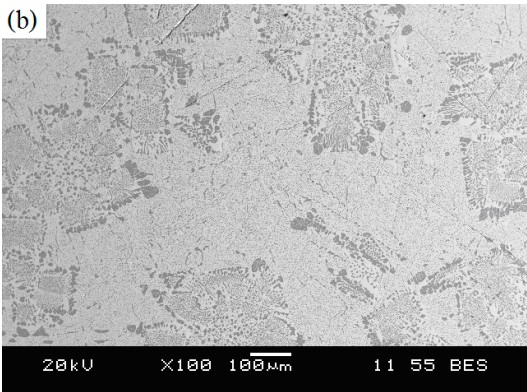

**Figure 7.** Microstructure of samples of Bi–18Sn–32Pb alloy (**a**) without treatment and (**b**) after EMP processing.

## 4. Discussion

### 4.1. Discussion of the Experimental Results

The observed complex structure of the fusion zone of the ingots of the Al–50Pb system can be explained by analyzing the state diagram of this system. In addition to the monotectic transformation, the indicated system was characterized by two more eutectic transformations at temperatures of 659 and 327 °C. Therefore, when the alloy of the selected composition was cooled to 659 °C, the mixture of two immiscible liquids (50.6% based on aluminum and 49.4% based on lead) underwent a eutectic transformation, decomposing into a mixture containing 98.6% aluminum crystals and 1.4% lead-based liquid phase. This liquid phase crystallized as globules in an aluminum matrix. During cooling, in the temperature range 659–327 °C, the number of aluminum crystals slightly increased, and aluminum was released at the globule-matrix interface. The liquid lead phase, which was not involved in the first eutectic transformation, was located in the lower layer of the ingot due to the difference in specific gravity and crystallized in the second eutectic transformation at 327 °C with the formation of a eutectic mixture of lead crystals of 99.24% and aluminum of 0.76%.

The movement of the lead phase with a lower crystallization temperature in the crystallized aluminum phase can be explained by the Tiller model [15]. According to his theory, the reason for this movement (thermomigration) is the difference in the phase equilibrium conditions along the interphase boundary. Under stationary thermal conditions (i.e., in the absence of temperature fluctuations), thermomigration can occur in the presence of a temperature gradient in a crystal containing a liquid inclusion. Due to the temperature dependence of the solubility of the crystal substance, warmer sections of the interface will dissolve, whereas less heated sections will crystallize. The resulting heterogeneity in the concentration of atoms of the substance of the crystal in the liquid phase will lead to the establishment of a constant flow of atoms of the substance of the crystal from more heated dissolving sections of the boundary to less heated crystallizing sections. This self-sustaining temperature gradient in the presence of a constant mass transfer determines the motion of the liquid inclusion in the crystal.

In addition, after EMP treatment, the concentration of lead at the fusion boundary decreased markedly from 82 to 41 wt %.

Calculation of the temperature dependences of the diffusion coefficients of aluminum and lead atoms [16] shows that by the time the mobility of aluminum atoms drops to almost zero, the mobility of lead atoms is still quite high. At crystallization/melting temperatures, the diffusion coefficients of aluminum and lead atoms differ by almost 15 times. This explains why the depth of the zone of separation of excess phases of lead atoms into aluminum is much greater than that of aluminum atoms into lead.

Because phases with a new crystal lattice do not form in the diffusion zone during diffusion, the observed process should be attributed to atomic diffusion. With this type of diffusion, the diffusion zone is a solid solution of a diffusing element in the crystal lattice of the solvent metal. Explanations of the observed effect should be on the basis of a well-known phenomenon called the Kirkendall effect [17], according to which if a pure metal and an alloy or two different alloys of two identical metals are kept at high temperature, the interface between them moves as a result of diffusion. Such experiments cannot be explained through direct or circular exchange of atoms, as in this case each movement of an atom in one direction should be accompanied by a movement of another atom in the opposite direction, that is, there should be no difference in the diffusion rate of the two components.

In terms of the EMP processing results of two- and three-component low-melting alloys, the obtained data on the quantitative phase ratio also indicated the occurrence of equilibrium crystallization in the EMP-treated metal. In the two-component alloy treated with EMP, the proportion of primary bismuth crystals was roughly 10%, and in the untreated alloy it was 50–55%. According to the equilibrium state diagram of the bismuth-lead system, the proportion of primary bismuth crystals in our alloy should be 13%. For the Bi-18Sn-32Pb three-component alloy, it was similar—after pulsed processing, the ratio of structural components was close to equilibrium. Analysis of the results showed that EMP processing had the greatest effect primarily in the early stages of crystallization, that is, changes the nature of nucleation. This was manifested in a change in the number of phases for a given alloy crystallizing at high temperatures. A decrease in porosity of the metal in both cases indicated a decrease in the number of casting defects in the metal being processed.

Summarizing the results of the above experiments, as well as the results of processing EMP alloys of other systems, the following general patterns of the influence of EMP on the properties of a metal can be noted: refining of structural elements, simultaneous increase in strength and ductility of the metal, increase in density, decrease in electrical resistivity, and increase in solubility of the main alloying elements in alpha phase [11–14]. Analysis of the revealed patterns allowed us to develop several options that explain the effect of EMPprocessing on the properties of metals.

### 4.2. Theories of the Mechanism of the Effect of EMP Processing on the Properties of Metals

Analysis of the presented and previous experiments allowed us to develop several theories about the effect of EMP on the properties of metals.

The following possible factors are considered as catalysts of changes in properties: electromagnetic stirring, thermal effect, and converting EMP energy to acoustic oscillation energy. Each of these factors can to one degree or another affect the structural state of the melt, causing changes in its crystallization characteristics. Let us analyze each of these possible factors.

#### 4.2.1. Electromagnetic Stirring

The contribution of this factor can be estimated by calculating the penetration depth of electromagnetic pulses into the melt (the so-called skin layer) and correlating it with the height of the melt processed in the crucible. In a previous study [18], we calculated the depth of the skin layer in a copper conductor for various durations and pulse shapes. For this, the strict electrodynamic problem of the propagation of a pulsed electromagnetic field in a conducting medium was solved. The spatiotemporal changes in the electromagnetic field in the depth of the electric current conductor, which were excited at the boundary of a solid body by external electric field strength, were calculated. The calculation results are shown in Table 2.

The table shows that the depth of the skin layer for the pulse that we used with a duration of 1 ns was measured in microns. For aluminum and iron, it would be 15–30% more. Such a thin layer cannot cause mixing of the entire mass of metal. Thus, it is obvious that electromagnetic stirring cannot have a significant effect on the properties of the metal during EMP processing of melts.

**Table 2.** Depth of electric field penetration into the conductor.

| Duration Pulse, s | The Depth of Penetration of the EMP into the Conductor, μm | |
| --- | --- | --- |
| | Rectangular Pulse | Pulse ($\sin^2 t$) |
| $10^{-3}$ | 5722 | 3576 |
| $10^{-6}$ | 181 | 113 |
| $10^{-9}$ | 5.7 | 3.6 |
| $10^{-12}$ | 0.18 | 0.11 |

### 4.2.2. Heat Exposure

Because the pulsed power was large and was measured in MW, even despite the short pulse duration, one can expect heating of the melt under electric pulse exposure, leading to a change in the structure and properties of the melt.

We calculated the thermal fields using computer simulation of pulsed exposure in a MathCad environment [19]. The thermophysical processes occurring during electric pulse exposure were described on the basis of reference data [20].

The calculation showed that during a pulse exposure of 1 ns with a power of 2 MW over the entire surface of the melt in the crucible, the depth of the heat-affected zone during the pulse exposure was 0.48 μm. The temperature of the melt on the surface during the pulse exposure increased by 16 °C and decreased almost to the initial value (an increase of 1.42 °C) at the depth of the heat-affected zone.

Thus, due to the insignificant size of the zone of thermal influence and temperature increase, the influence of the thermal factor during EMP processing can also be neglected.

### 4.2.3. Converting Electromagnetic Pulses to Acoustic Pulses

Previous research [21–24] has shown that ultrasonic vibrations occur in metal melts when exposed to electromagnetic waves. In [24], possible schemes for the electrodynamic induction excitation of vibrations in liquid metals were classified, and the ultrasound intensity in the metal was estimated, which can be obtained with parameters close to real parameters. The contactless excitation scheme of elastic vibrations in a melt using a constant magnetic field and alternating current was considered in detail. The calculation performed for induction melting of aluminum showed that when aluminum was melted in a crucible with a diameter of about 300 mm with a constant field of $5 \times 10^4$ A/m, the vibrational pressure on the melt was 2 atm. It was assumed that this pressure was sufficient in obtaining useful metallurgical effects. The occurrence of mechanical vibrations in metal samples is possible without applying an external magnetic field [25]. A similar situation is also characteristic of metal melts [25,26].

Because the effects observed during the EMP processing of melts were similar to those observed during ultrasonic processing, we can conclude that the melt also experienced pressure from the acoustic field generated by the EMP generator. Due to the design feature of the installation used, the actual excitation of mechanical vibrations occurred by contact method. Due to the arrangement of the radiator, current flows from the radiator along the surface of the melt and, possibly, the efficiency of the excited oscillations was much greater than in the aforementioned research.

To theoretically substantiate the given hypothesis about the mechanism of influence of EMP on metal melts, we carried out a comparative calculation of the vibration intensity. Sound pressure (*p*) and particle displacement in metals for both plane and spherical waves are related by the relationship [21]:

$$p = \rho v \omega \xi = z \omega \xi \qquad (1)$$

where the product of the metal density ($\rho$, kg/cm$^3$) and the speed of sound ($v$, m/s): $\rho v = z$ acoustic impedance (resistance); $\omega$ is the circular frequency ($\omega = 2\pi f$; Hz); and $\xi$ is the displacement of particles from the equilibrium position (μm).

Using this formula, we can examine the parameters provided by the authors of existing research on melt treatment and calculate the sound pressure in their experiments. In [27], for a radiator in an aluminum alloy: $f$ = 21 kHz, $\xi$ = 25 μm, generator power 1 kW, vibration intensity 100 W/cm$^2$; mass of the processed metal was not indicated. Longitudinal wave velocity in solid aluminum was found to be 6260 m/s [28]. The oscillation velocity in the melt can be 70% of the velocity in a solid metal [28]. The density of molten aluminum is 2390 kg/m$^3$ [29]. Utilizing Equation (1) above, we obtained a sound pressure of 29 MPa or 290 atm.

In [30], for a radiator in an aluminum alloy, the authors provided the following parameters: $f$ = 20 kHz, $\xi$ = 4 μm, generator power 150 W, mass of the metal being processed was 200 g. Calculation of sound pressure gave a value of 4.6 MPa (46 atm).

In [31], for a radiator in an aluminum alloy, $f$ = 19.5 kHz, $\xi$ = 30 μm, generator power of 600 W, oscillation intensity of 109 W/cm$^2$, mass of the processed metal was 210 g. Calculation of sound pressure gave a value of 34 MPa (340 atm).

In [24], it was noted that in order to create positive metallurgical effects, the electrodynamic pressure in the melt should be 1–4 atm or 1–4 $\times$ 10$^5$ Pa.

To determine the vibrational pressure from the impact of EMP, the following equation can be used for calculating the wave pressure on the surface:

$$p = E(1 + R)/c, \tag{2}$$

where p is wave pressure, N/m$^2$; E is the incident wave power, on the unit area and in the unit time, W/cm$^2$; $R$ is the reflection coefficient ($R$ = 0 at full absorption, $R$ = 1 at full reflection); and $c$ is the wave propagation velocity, m/s. The wave propagation velocity in molten metals is in the order of $4 \times 10^3$ m/s [28].

With pulsed excitation of oscillations, the incident pulsed power can be approximately calculated by the following equation:

$$P = \frac{U^2}{r}, \tag{3}$$

where $r$ is cable impedance of 50 Ohms and $U$ is 10 kV generator voltage. Having calculated by Equation (3) the value of the incident pulse power, we obtained P = $2 \times 10^6$ W. The free surface area of the metal in the crucible with a diameter of 80 mm was $5 \times 10^{-3}$ m$^2$. Therefore, the pulse power per unit area was found to be $4 \cdot \times 10^8$ W/m$^2$.

Substituting the obtained value of the pulse power in Equation (1), we obtained $P_{pulse}$ = $1.3 \times 10^5$ Pa (or 1.3 atm). This value is close in magnitude to the values of pressure used in [24,30].

## 5. Conclusions

Both the results of our previous studies and the proposed hypothesis on the mechanism of the influence of EMP on the properties of metals showed that this treatment had the same effect on the properties of metals as ultrasonic treatment. However, our technology had a number of indisputable advantages: power consumption of the generator was only 100 W, and EMP processing was effective not only for small volumes of metal, but also for melts weighing up to 3 tons, whereas ultrasound treatment of such volumes is inefficient.

According to the results of our studies, the following conclusions can be drawn:

1. EMP processing affected the crystallization process—it brought crystallization close to equilibrium and affected the diffusion processes in the metal.
2. After EMP processing, the phase composition of alloys remained unchanged, but the proportion of phases in the structure changed significantly.
3. The density of the EMP-processed alloys was found to be slightly higher than the alloys without pre-treatment.
4. The most likely model for the influence of EMP on the properties of the metals was the occurrence of acoustic pulses in the melt during their processing. The calculation results showed that the

pressure created by acoustic pulses was commensurate with the pressure arising from ultrasonic action. Due to the periodicity of the appearance of acoustic pulses and the high pressure created by them, they can cause changes in the metal.

**Author Contributions:** Conceptualization, V.K. and N.S. methodology, V.K. and N.S.; validation, V.K., N.S., and E.L.; formal analysis, V.K.; investigation, N.S.; writing—original draft preparation, N.S. and E.L.; writing—review and editing, V.K., N.S., and E.L.; visualization, E.L.; supervision, V.K.; project administration, V.K. All authors have read and agreed to the published version of the manuscript.

**Funding:** This research received no external funding.

**Conflicts of Interest:** The authors declare no conflict of interest.

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
