# Peer review of "Analysis of the Results of Pulsed Processing of Melts"

_metals, doi:10.3390/met10020205_

Round 1
Reviewer 1 Report
This is an improved version of the manuscript that was submitted earlier. I think there is evidence of some effects of the EMP treatment, but in lines 234-239, there are claims relating to "refining of structural elements, simultaneous increase in strength and ductility of the metal, increase in density, decrease in electrical resistivity, and increase in solubility of the main alloying elements in alpha phase." There is evidence of grain refinement in one of the compositions, however, the rest of the claims are not backed up by any data so all this appears to be speculation. As a result, the significant of the content is low as these claims seem speculative. More data are needed to back theses claims and to explain them. For this reason, the manuscript cannot be accepted in its current form. Effectively stronger evidence is required. In addition the following need to be addressed:
line 83, the previous studies on which the duration is determined need to be referenced. In lines 110-112, how were the amounts measured? In line 149, you mean "equilibrium phase diagram." In lines 165-166, and 232, it is better to say that there was a decrease in the porosity levels and also present porosity data (rather than talk about a density increase as this can be misunderstood). An explanation of this will be useful. There was an explanation in the response of the authors. It is better to use the term "nucleation sites" and not "germinal centres." As in the comments above in lines 234-239, the claims are not backed up with measurements.Author Response
Dear Reviewer, first of all, we want to thank you for your attention and interest in our work. And let me answer your comments.
This is an improved version of the manuscript that was submitted earlier. I think there is evidence of some effects of the EMP treatment, but in lines 234-239, there are claims relating to "refining of structural elements, simultaneous increase in strength and ductility of the metal, increase in density, decrease in electrical resistivity, and increase in solubility of the main alloying elements in alpha phase." There is evidence of grain refinement in one of the compositions, however, the rest of the claims are not backed up by any data so all this appears to be speculation. As a result, the significant of the content is low as these claims seem speculative. More data are needed to back theses claims and to explain them. For this reason, the manuscript cannot be accepted in its current form. Effectively stronger evidence is required.
Answer.
We completely agree with you. Some of our statements may seem unfounded. However, the volume of the manuscript does not allow us to cite the results of all our studies. About the properties changes you mentioned, we have repeatedly written in earlier publications. However, most of them are published in Russian-language literature and are inaccessible to a wide circle of readers. But some results are available. So in [11 - Krymsky, V., Shaburova, N. Applying of pulsed electromagnetic processing of melts in laboratory and industrial conditions. Materials 2018, 11 (6), 954. DOI: 10.3390 / ma11060954.] on pages 5-8, we present the results of resizing microstructural components and mechanical properties of aluminum alloys. In addition, for review, we send you the results on Pulse processing of steels (from the section of our book Balakirev, VF; Krymsky, VV; Ri, H .; Shaburova, NA Electric Pulse Treatment of Metal Melts; UB RAS: Ekaterinburg, Russia, 2014; 144p, ISBN 978-5-87184-640-7). We also added a link to the book to the text.
“Balakirev, V.F.; Krymsky, V.V.; Ri, H.; Shaburova, N.A. Electric Pulse Treatment of Metal Melts; UB RAS: Ekaterinburg, Russia, 2014; 144p, ISBN 978-5-87184-640-7.
Chapter 2.6. EMP processing of low carbon steels.
Steel processing was carried out in workshop conditions at Uralvagonzavod OJSC. The chemical composition of the studied steels corresponds to GOST 977-88 (steel 20L), OST 32.183-2001 (steel 20GL).
Two types of parts were processed No. 132.01.02.035-0 (ad-hoc) square No. 68431 and No. 100.10.014-0 (axle box body) square No. 698431. Details were filled in by a bush method of 6 pcs. in the bush.
Processing method: a pulsed electromagnetic field is fed into a mold with molten metal through two electrodes. The electrodes in the mold were positioned so that between them was the largest mass of metal. The processing time from the beginning of pouring the metal into the mold for the elevators is 5-10 minutes, for the axle box body - 20 minutes.
The analysis of the mechanical properties and structure of the embedded samples was carried out. Both samples are characterized by ferrite-pearlite structure. The grain size in samples processed by EMP is 8, 9, in untreated - 8.
The mechanical properties of steel samples 20L were determined during tensile tests. The test results are given in table. 2.5.
Table 2.5. Mechanical properties of steel 20L
|
Sample |
σв, MPа |
σ0.2, MPа |
δ, % |
ψ, % |
KCU–60 °C, J/cm2 |
|
Without EMP processing |
291 |
523 |
26 |
39 |
910–1040 |
|
EMP processed |
314 |
511 |
33 |
53 |
910–1090 |
Thus, in general, one can note an increase in both the strength properties and the ductility properties.
Similar studies were conducted for steel 20GL. In the table. 2.6 shows the mechanical properties of samples of steel 20GL. The microstructure of steel samples 20GL is shown in Fig. 2.20, 2.21 (a - microstructure of the surface, b - microstructure of the middle of the part).
Table 2.6 Mechanical properties of steel 20GL
|
Sample |
σв, MPа |
σ0.2, MPа |
δ, % |
ψ, % |
KCU–60 °C, J/cm2 |
|
Without EMP processing |
507 |
296 |
24 |
42 |
17–20 |
|
EMP processed |
552 |
336 |
28,5 |
52 |
34–25 |
|
|
|
а |
б |
|
Fig. 2.20. Microstructure of a part made of steel 20GL without EMP treatment |
|
|
а |
б |
|
Fig. 2.21. The microstructure of the part made of steel 20GL treated with EMP |
|
Chapter. 2.7. Steel processing 35L
Irradiation of the metal was carried out before casting in a steel casting ladle with a diameter of 800 mm and a height of 1200 mm. Mass of irradiated metal: 1.9 tons.
Both samples are characterized by the presence of a ferrite-pearlite structure. In this case, the ferrite phase is present as a grid along the boundaries of pearlite grains. The microstructure of the samples after etching is shown in Fig. 2.23.
|
а |
b |
|
Fig. 2.23. Etched thin sections: a - initial sample, b - processed EMP |
|
The presence of precipitates of excess ferrite at the grain boundaries allows us to determine the average grain size. So in the initial sample it is 700 μm, in the EMP-treated 450 μm.
The results of mechanical tensile and bending tests are given in table 2.8.
Table 2.8. The results of mechanical tests of steel 35L
|
Sample |
σв, MPа |
σ0.2, MPа |
δ, % |
ψ, % |
KCU–60 °C, J/cm2 |
|
Without EMP processing |
386 |
520 |
6 |
12 |
9,8 2,5 |
|
EMP processed |
454 |
772 |
20 |
39 |
13 |
line 83, the previous studies on which the duration is determined need to be referenced.
Answer. Thanks for the comment, we added the link.
In lines 110-112, how were the amounts measured?
Answer. Estimation of the number of structural components was carried out by means of the hardware-software complex of image analysis Thixomet Pro.
In line 149, you mean "equilibrium phase diagram."
Answer. Yes, thanks for the comment, that is what we had in mind.
In lines 165-166, and 232, it is better to say that there was a decrease in the porosity levels and also present porosity data (rather than talk about a density increase as this can be misunderstood). An explanation of this will be useful. There was an explanation in the response of the authors. It is better to use the term "nucleation sites" and not "germinal centres." As in the comments above in
Answer. Yes, thanks, you're right. We have made changes to the text.
lines 234-239, the claims are not backed up with measurements.
Answer. Yes, thanks, you're right. We have added links to previous publications.
Once again, we want to thank you for the valuable comments that will certainly improve the quality of our future research.
Also, if our research interests you, we are ready to send you an electronic version of our books. But, unfortunately, in the Russian version.
Respectfully,
Team of Authors.

Reviewer 2 Report
A very interesting experimental paper that details very novel experiments. However, I have some concerns with the theoretical explanations presented.
More detail is needed on how the skin depth in the “Electromagnetic stirring” section (Lines 247-257). Though the authors have referenced a previous paper of theirs it seems to be difficult to find online to check the methodology. More detail on how the model was setup in MathCad (line 261) would be beneficial. I am not convinced by their assumptions in their ultrasonic explanation. It may be possible for cavitation to be occurring in the melt however they need to do more to confirm this. Ultrasonic vibrations on their own have been shown to be insufficient without cavitation. Cavitation occurring is not just a function of pressure, frequency is also important, what frequency of acoustic waves to the authors predict? The equation on line 313 needs either a reference or a derivation. Cavitation only occurs above the Blake threshold for 5-micron radius bubbles this is 2.5 kPa in aluminium melts An estimate of 4000 m/s is used for the speed of sound, though this is approximately correct for molten Aluminium (4560m/s) [1], it is a lot lower for Bismuth and Lead which are also both used. A Bi-Pb alloy is likely around 1700 m/s. (Pb is 1821 and Bi 1640)[1] Sound velocity of liquid metals and metalloids at the melting temperature, Physics and Chemistry of Liquids: An International Journal, 45:4, 399-407
Author Response
Dear Reviewer, first of all, we want to thank you for your attention and interest in our work. And let me answer your comments.
More detail is needed on how the skin depth in the “Electromagnetic stirring” section (Lines 247-257).
Answer. To determine the depth of the skin layer, the strict electrodynamic problem of the propagation of a pulsed electromagnetic field in a conducting medium was solved. The spatiotemporal changes in the electromagnetic field in the depth of the electric current conductor, which were excited at the boundary of a solid body by external electric field strength, were calculated. The calculation results are shown in the manuscript in table 2.
Though the authors have referenced a previous paper of theirs it seems to be difficult to find online to check the methodology. More detail on how the model was setup in MathCad (line 261) would be beneficial.
Answer. The MathCad was used to calculate the equations of temperature distribution in the surface layer of the melt. For the calculation, standard equations were used.
I am not convinced by their assumptions in their ultrasonic explanation. It may be possible for cavitation to be occurring in the melt however they need to do more to confirm this. Ultrasonic vibrations on their own have been shown to be insufficient without cavitation. Cavitation occurring is not just a function of pressure, frequency is also important, what frequency of acoustic waves to the authors predict? The equation on line 313 needs either a reference or a derivation. Cavitation only occurs above the Blake threshold for 5-micron radius bubbles this is 2.5 kPa in aluminium melts An estimate of 4000 m/s is used for the speed of sound, though this is approximately correct for molten Aluminium (4560m/s) [1], it is a lot lower for Bismuth and Lead which are also both used. A Bi-Pb alloy is likely around 1700 m/s. (Pb is 1821 and Bi 1640)
[1] Sound velocity of liquid metals and metalloids at the melting temperature, Physics and Chemistry of Liquids: An International Journal, 45:4, 399-407
Answer. We consider the following comments necessary for this comment. Usually, ultrasonic generators operate on sinusoidal oscillations of a certain frequency. For the pulses we use, the concept of frequency is unacceptable. In theory, an impulse can be represented as an expansion in a Fourier series (line spectrum). Linear spectrum: on the frequency scale, the lines are located through 1 kHz (pulse repetition rate) from zero to 109 Hz (which is the inverse of the pulse duration) with decreasing amplitude. But this is in theory. But in fact there is an acoustic impulse (shock) lasting 10-9 seconds.
Due to the short pulse duration (10–9 sec), conditions for the occurrence of cavitation in such a dense medium as a metal melt are hardly possible. Because after the pulse there follows a long period of time (10-3 seconds) during which no effect occurs. Those. powerful but rare impacts occur that cause melt mechanical vibrations and affect crystallization processes.
The calculation of the arising pressure in the manuscript, as indicated, is carried out on the example of aluminum alloys. Because It is for these materials in the literature that there are many results on the assessment of pressure arising from ultrasonic treatment. And it is indicated that a pressure of this level is sufficient to influence the structure and properties of the metal. In the calculations, we make an approximate estimate, therefore, we consider some inaccuracies (the speed of sound in aluminum 4000 m / s, rather than 4560 m / s) acceptable. For lead and bismuth, we did not make calculations, because there is no data in the literature on their processing of ultrasonic vibrations which could be used for comparison.
Respectfully,
Team of Authors.

Reviewer 3 Report
Dear Authors,
The paper presents interesting results based on the analysis pulsed processing of melts. The presented methodology is relatively new and original. As a Reviewer, I suggested MINOR revision. Before publication, please consider the points listed d below:
1) Page 2 line 74 - is watts should be Watts
2) Page 7 line 189 - it would be better to present temperature range from minimal up to maximal value. However, it is clear that is the cooling process.
3) Page 9 - Eq. 1 - Not all symbols are mentioned and defined in the text.
4) Page 9 line 294 - it is suggested to change the symbol of the speed of sound (is c), because c is well known in physics as light speed.
5) Sec. 4.2.3 - I am not sure that the title is adequate. Mechanical Impact suggest that we can expect more mechanical contribution. Please consider the rename of it or it would be great to add more results - much stronger connected with the original title.
6)Also, it is recommended to underline the statistical analysis in order to better validation. In the presented paper, it is a strong lack of it...
Best regards
Rev.
Author Response
Dear Reviewer, first of all, we want to thank you for your attention and interest in our work. And let me answer your comments.
Page 2 line 74 - is watts should be Watts
Answer. Thank you for your comment. We have made changes to the text of the manuscript.
Page 7 line 189 - it would be better to present temperature range from minimal up to maximal value. However, it is clear that is the cooling process.
Answer. Yes, You are right. We purposefully indicated the temperature range from maximum to minimum, so that the reader could imagine that it is in this interval that the transformation occurs during cooling.
Page 9 - Eq. 1 - Not all symbols are mentioned and defined in the text.
Answer. Thank you, we checked and made changes to the text.
Page 9 line 294 - it is suggested to change the symbol of the speed of sound (is c), because c is well known in physics as light speed.
Answer. Thank you, we checked and made changes to the text.
5) Sec. 4.2.3 - I am not sure that the title is adequate. Mechanical Impact suggest that we can expect more mechanical contribution. Please consider the rename of it or it would be great to add more results - much stronger connected with the original title.
Answer. We agree with your comment. The name does not fully match the content. We have changed the title.
6) Also, it is recommended to underline the statistical analysis in order to better validation. In the presented paper, it is a strong lack of it...
Answer. The paper presents the averaged results for three experiments for each group of alloys. The scatter of the obtained values in the experiments was not significant.
Respectfully,
Team of Authors.

Round 2
Reviewer 1 Report
The discussion provides results and analysis from earlier published work. The analysis is not for the current alloy systems that were investigated in the manuscript, but for systems that were studied previously. Thus the critical part of the discussion is based on results that have already been published elsewhere. This diminishes the originality of the work as the analysis provided is not new, but has already been published elsewhere. For this reason the manuscript is rejected.
Author Response
Dear Reviewer.
We express our deep gratitude for the interest shown in our work. Your comments are very important to us, and not only within the framework of this manuscript, but also when planning further research.
Let us answer your last comment:
“The discussion provides results and analysis from earlier published work. The analysis is not for the current alloy systems that were investigated in the manuscript, but for systems that were studied previously. Thus the critical part of the discussion is based on results that have already been published elsewhere. This diminishes the originality of the work as the analysis provided is not new, but has already been published elsewhere. For this reason the manuscript is rejected.”
Answer:
A discussion of the results of the studies is given in the section “4.1. Discussion of the experimental results» of the manuscript. At the end of this section, to the results obtained on Al-Pb, Bi-Pb, Bi-Sn-Pb alloys, we added (overview) the results of previous studies obtained on other systems. They are necessary for the full justification of our proposed models of the impact of EMR on the properties of metal.
The results presented in the manuscript are new, not previously published. The results of previous studies that we mentioned were published only in Russian-language publications and are inaccessible to a wide range of readers. Therefore, we consider it necessary to mention them.
Reviewer 2 Report
Author Answer: We consider the following comments necessary for this comment. Usually, ultrasonic generators operate on sinusoidal oscillations of a certain frequency. For the pulses we use, the concept of frequency is unacceptable. In theory, an impulse can be represented as an expansion in a Fourier series (line spectrum). Linear spectrum: on the frequency scale, the lines are located through 1 kHz (pulse repetition rate) from zero to 109 Hz (which is the inverse of the pulse duration) with decreasing amplitude. But this is in theory. But in fact there is an acoustic impulse (shock) lasting 10-9 seconds.
Due to the short pulse duration (10–9 sec), conditions for the occurrence of cavitation in such a dense medium as a metal melt are hardly possible. Because after the pulse there follows a long period of time (10-3 seconds) during which no effect occurs. Those. powerful but rare impacts occur that cause melt mechanical vibrations and affect crystallization processes.
While there may be acoustic waves generated by the EMP, the authors acknowledge that there is no cavitation. Acoustic cavitation is generally accepted to be the mechanism by which ultrasonic treatment results in grain refinement. If they are proposing an alternative mechanism, they should detail this.
Author Response
Dear Reviewer.
We express our deep gratitude for the interest shown in our work. Your comments are very important to us, and not only within the framework of this manuscript, but also when planning further research.
Let us answer your last comment:
«Author Answer: We consider the following comments necessary for this comment. Usually, ultrasonic generators operate on sinusoidal oscillations of a certain frequency. For the pulses we use, the concept of frequency is unacceptable. In theory, an impulse can be represented as an expansion in a Fourier series (line spectrum). Linear spectrum: on the frequency scale, the lines are located through 1 kHz (pulse repetition rate) from zero to 109 Hz (which is the inverse of the pulse duration) with decreasing amplitude. But this is in theory. But in fact there is an acoustic impulse (shock) lasting 10-9 seconds.
Due to the short pulse duration (10–9 sec), conditions for the occurrence of cavitation in such a dense medium as a metal melt are hardly possible. Because after the pulse there follows a long period of time (10-3 seconds) during which no effect occurs. Those. powerful but rare impacts occur that cause melt mechanical vibrations and affect crystallization processes.
While there may be acoustic waves generated by the EMP, the authors acknowledge that there is no cavitation. Acoustic cavitation is generally accepted to be the mechanism by which ultrasonic treatment results in grain refinement. If they are proposing an alternative mechanism, they should detail this.»
Answer.
At this stage of the research, we cannot clearly confirm the presence or absence of cavitation. We do not have the appropriate equipment. However, our calculations show that when pulsed by EMR in a metal, acoustic pulses arise that have a pressure comparable to that arising from ultrasonic treatment. In our opinion, not only the magnitude of their pressure, but also the frequency is important. In fact, during EMR processing, shock waves periodically arise, which affect crystallization processes (they crush crystallization centers, etc.). Cavitation processes during ultrasonic processing act similarly, but the nature of their action is continuous.
We have made additions to the section "5 Conclusion".
This manuscript is a resubmission of an earlier submission. The following is a list of the peer review reports and author responses from that submission.
Round 1
Reviewer 1 Report
The focus of the present document is to present the effect of Electromagnetic Pulse in the Metal Melts processing. For that, the authors are using a set of considerations based on theoretical concepts in order to justify the benefits of this technique of processing. Although the document is well established and organized, I consider that there are several remarks to address before to be published.
(1) Although there are several documents regarding this issue, them were not mentioned in Introduction. A better introduction regarding the effect of mechanical/physical processing of metal melts should be given.
(2) I don’t know if the sentence of line 49 and 50 is correct. There are several works already published about this issue.
(3) What means “Fusible alloys were selected for the following reasons.” – Line 54. Fusible alloys? It is suggested to use a standard denomination for such kind of process.
(3) The dimensions of the crucible to melt the alloys are the same for all castings? A better description of the methodology used should be introduced in document, e.g. temperatures.
(4) Although in Table 1 are presented 4 different alloys, in Section of results just some of them are described.
(5) In a work where the main focus is to show the effect of the EMP in the microstructure of the alloys, the results present in Figure 2, 3 and 4 is not the most convenient. A different approach is suggested, e.g. use of OM and SEM/EDS technique. Size of secondary phases after EMP processing? Grain Size? Morphology of the phases.
(6) Figure 5 is not clear. Concentration vs Distance?
(7) Although the authors introduced a wide discussion of the results, it is based on theoretical approach which is not totally understood yet. For instance, I have some reserves regarding the justifications presented in 4.2.3 Mechanical Impact subchapter. Comparing EMP and ultrasonic processes is not totally correct. It is suggested to make the discussion of results based on the complete analyses of results to be inserted in Results chapter.
Author Response
First of all, let us thank you for carefully reading our work and for criticizing us.
Also let me make the following comments.
(1) Although there are several documents regarding this issue, them were not mentioned in Introduction. A better introduction regarding the effect of mechanical/physical processing of metal melts should be given.
Answer.
Thank you for your comment. We have made appropriate changes to the text of the manuscript.
(2) I don’t know if the sentence of line 49 and 50 is correct. There are several works already published about this issue.
Answer.
You are right, many researchers, in particular, those authors whose links we added in accordance with your first remark, are studies of the influence of electromagnetic effects on melts. However, the uniqueness of our technique lies in the equipment used. Our generator generates electromagnetic pulses with unique characteristics. First of all, it is: pulse duration 1 ns, pulse amplitude 10 kV, rated impulse power 2 MW and very low power consumption. Initially, such generators were used to solve radar problems. And only dr. Krymsky V. in 2001 for the first time used this generator to act on metal melts. Therefore, if you met the work about this particular method of exposure, it was the work of dr. Krymsky V. or his laboratory staff.
(3) What means “Fusible alloys were selected for the following reasons.” – Line 54. Fusible alloys? It is suggested to use a standard denomination for such kind of process.
Answer.
Thank you for your comment. We have made appropriate changes to the text of the manuscript.
(3) The dimensions of the crucible to melt the alloys are the same for all castings? A better description of the methodology used should be introduced in document, e.g. temperatures.
Answer.
Thank you for your comment. We have made appropriate changes to the text of the manuscript.
(4) Although in Table 1 are presented 4 different alloys, in Section of results just some of them are described.
Answer.
Thank you for your comment. Indeed, 4 alloys are indicated in Table 1, however, one of them is mistakenly re-coping. We corrected it.
(5) In a work where the main focus is to show the effect of the EMP in the microstructure of the alloys, the results present in Figure 2, 3 and 4 is not the most convenient. A different approach is suggested, e.g. use of OM and SEM/EDS technique. Size of secondary phases after EMP processing? Grain Size? Morphology of the phases.
Answer.
Thank you for your comment. We have made appropriate changes to the text of the manuscript.
(6) Figure 5 is not clear. Concentration vs Distance?
Answer.
To plot this graph, first on a SEM equipped with an attachment for XRMA, scanning along the line in the fusion zone of two parts of the ingot was made and the radiation intensity of Al and Pb was determined (see Fig. 4c and d). After that, using the Grafula program, the results were digitized and a graph was plotted for the distribution of the number of elements (in mass%) depending on the distance near the fusion zone. It is this graph that is presented in Fig. 5.
(7) Although the authors introduced a wide discussion of the results, it is based on theoretical approach which is not totally understood yet. For instance, I have some reserves regarding the justifications presented in 4.2.3 Mechanical Impact subchapter. Comparing EMP and ultrasonic processes is not totally correct. It is suggested to make the discussion of results based on the complete analyses of results to be inserted in Results chapter.
Answer.
Yes, you are absolutely right. We agree that our theory is not yet fully developed. But this is the initial stage, and so far we are claiming only a hypothesis. But we consider it necessary to clarify the correctness of comparing EMP and ultrasound processing. According to the existing laws of physics, the electromagnetic pulses we use cannot penetrate into the melt deeper than 1-1.5 mm (we are talking about the skin-layer). However, as shown by numerous experiments, the effect of such processing is present . At the same time, there are works (for example, S. Gurevich) that prove that the excitation of ultrasound in both melts and solid metal is possible under the influence of an electromagnetic field. We cannot directly fix the occurrence of ultrasound in our experiments, because The EMP generator creates large interference for the operation of electronic equipment near it. But based on a comparison of the effects produced by EMP and ultrasonic treatment on alloys similar to our compositions, we were convinced that the occurrence of ultrasound in our conditions is real.
Reviewer 2 Report
The presented paper is focussing on the investigation of the effect of electromagnetic pulses on the solidification behaviour and resulting solid structure of several alloys (Al-Pb, Bi-Pb and Bi-Sn-Pb). The paper starts with the presentation of earlier results in this field, mainly of the authors.
The reviewer recommends this papers to be rejected, as - in his view - it has several significant flaws as described in the following.
1.) The motivation (line 56ff) for the research work is absolutely not clear: how can EMP and soldering in microelectronics be combined? Further point of the authors is the “development of a theory” to explain the effects of EMP. However, latter theory is rather vague and scientifically not sound (see below).So, in conclusion: neither "application-related" nor "fundamental" scientific aspects are adressed in a qulity and soundness which would be interesting for the reader.
2.) The impression of the reviewer is that the authors used the existing technique just for another group of alloys.
3.) The quality of the text/language makes it difficult to follow discussion of effects in the system Al-Pb. What is the significance of the layer structure for any technical process?
4.) The proposed "theory" should cover (maybe not in detail but at least in general) the whole way from the EMP to its mechanical/thermal... effects to the influencing of melt structure and solidification path and finally to the resulting solid structure. This was not done.
5.) Diffusion effects are considered in the discussion of the origin of the differences in the boundary layer. No information about cooling rate in the copper mould is given. So, how much time is available for the diffusion processes to take place? The authors write a lot about diffusion effects (Tiller model, Kirkendall effect, etc.) but it is not clear how the EMP will affect those.
6.) Text structure: sub-chapter 4.1 is called “discussion of experimental results”, but the content is only on one system (Al-Pb) and does not really explain the observed boundary structure and the role of EMP on it. Sub-chapter 4.2. is focussing on the pressure impacts and comparison to ultra-sonic treatments, but the connection of EMP and the alloy structure is explained rather vaguely by “equilibrium crystallisation”.
7.) Thermoanalysis is often used when melt treatments are investigated (undercooling etc.). Why did the authors not include such important additional data?
8.) The discussion of the Bi-Pb-system is also not detailed enough. How does EMP lead to the solidification according to the equilibrium?
When starting reading the text the reviewer noted several less important aspects to be improved. However, when he considered to recommend "reject" he stopped this. Anyway, those remarks are listed in the following
Detailed remarks
-quality of metallographic figures (polishing!) is not good.
-line 20 : melt treatment out? Unclear expression
-line 28: degassing and uniform ….
-line 30: When shaking… revise sentence
-line 34: was tested
-line 40: and an increase
-line 42: revise sentence
-line 46: 35L seems to be a Russian steel designation. An international code should be added.
-line 53: the alloys differ significantly in their behaviour. It should at least be mentioned in this part of the text that Al-Pb is monotectic and shows a miscibility gap also in the melt
-line 57: (it was important…): the connection of soldering and the processing of large melt batches with EMP seems not really straightforward and should be explained. Electromagnetic pulses and microelectronics go not really well together.
-line 76: two melts were carried out: revise expression
-line209: Kirkendall
Author Response
First of all, let us thank you for carefully reading our work and for criticizing us.
Also let me make the following comments.
1.) The motivation (line 56ff) for the research work is absolutely not clear: how can EMP and soldering in microelectronics be combined? Further point of the authors is the “development of a theory” to explain the effects of EMP. However, latter theory is rather vague and scientifically not sound (see below).So, in conclusion: neither "application-related" nor "fundamental" scientific aspects are adressed in a qulity and soundness which would be interesting for the reader.
Answer.
Thank you for your comment. We made changes to the text of the manuscript on the motivation of the study and in the conclusion section.
Regarding the combination of soldering and EMP we can say the following. Our work is not about a combination of pulsed processing and soldering processes. We describe a method for manufacturing an alloy that can be used for soldering.
2.) The impression of the reviewer is that the authors used the existing technique just for another group of alloys.
Answer.
Thanks for the question. As noted in the manuscript, only two laboratories in the world deal with the issues of the impact of EMR with similar characteristics on the properties of various materials. Because Initially, these pulse generators were used for radar. And, as indicated in the manuscript, before starting to develop a theory about the mechanism of the effect of EMP treatment on the properties of metals, it was necessary to conduct numerous studies that would allow accumulating sufficient statistical material. Which we did.
3.) The quality of the text/language makes it difficult to follow discussion of effects in the system Al-Pb. What is the significance of the layer structure for any technical process?
Answer.
Thank you for your comment. Perhaps difficulties are associated with the quality of the translation of the manuscript. We will try to fix it. On the second part of the question: in some electrical devices (for example, in high-capacity batteries) it is better to have external terminals made of aluminum or copper, as the existing connection methods at high currents are inefficient.
4.) The proposed "theory" should cover (maybe not in detail but at least in general) the whole way from the EMP to its mechanical/thermal... effects to the influencing of melt structure and solidification path and finally to the resulting solid structure. This was not done.
Answer.
We fully agree that the theory we are proposing is not quite complete. Now we are considering only hypotheses. Since the development of a rigorous mathematical theory requires not only time, but also financial means.
In this work the mechanical characteristics were not really paid attention to, since alloys of the studied systems are not used as structural alloys. As for the integrity of the theory "from the EMP to its mechanical/thermal... effects to the influencing of melt structure and solidification path and finally to the resulting solid structure", the following may be noted. According to the theory that we propose, under the influence of electromagnetic radiation, ultrasonic vibrations arise in the melt. The calculations performed in the work show that the sound pressure level during EMR exposure is close to the sound pressure that occurs during ultrasonic treatment, despite the fact that the natures of the treatments are different. The mechanism of the influence of ultrasound on the mechanical characteristics is described in the literature.
5.) Diffusion effects are considered in the discussion of the origin of the differences in the boundary layer. No information about cooling rate in the copper mould is given. So, how much time is available for the diffusion processes to take place? The authors write a lot about diffusion effects (Tiller model, Kirkendall effect, etc.) but it is not clear how the EMP will affect those.
Answer.
Thank you for your comment. We have made changes to the text of the manuscript.
We used the Tiller and Kirkendall model to explain the structures observed in a raw metal (without EMP treatment). In particular, the appearance of a lead phase in the aluminum part of the ingot. Since similar phenomena are also observed in the EMP-treated metal, it is obvious that EMP does not significantly affect these effects
6.) Text structure: sub-chapter 4.1 is called “discussion of experimental results”, but the content is only on one system (Al-Pb) and does not really explain the observed boundary structure and the role of EMP on it. Sub-chapter 4.2. is focussing on the pressure impacts and comparison to ultra-sonic treatments, but the connection of EMP and the alloy structure is explained rather vaguely by “equilibrium crystallisation”.
Answer.
Thanks for the comment, we completely agree with you and we have made changes to the text of the manuscript. We also want to note that in order to establish a clear relationship between the EMP treatment and the effects produced by this treatment on the metal, detailed studies of the intermediate states of the melt and the metal at the crystallization stage are needed. At this stage of the work, in view of the wide variety of materials studied, we were only interested in the final results of such an impact.
7.) Thermoanalysis is often used when melt treatments are investigated (undercooling etc.). Why did the authors not include such important additional data?
Answer.
Thanks for the recommendation for using thermal analysis. In this work, we did not have the necessary equipment. But further research planned to carry out such work.
8.) The discussion of the Bi-Pb-system is also not detailed enough. How does EMP lead to the solidification according to the equilibrium?
Answer.
Thank you for your comment. We have made changes to the text of the article.
And thank you for detailed remarks.
-line 20 : melt treatment out? Unclear expression
We corrected it.
-line 28: degassing and uniform ….
We corrected it.
-line 30: When shaking… revise sentence
We corrected it.
-line 34: was tested
We corrected it.
-line 40: and an increase
We corrected it.
-line 42: revise sentence-line 46: 35L seems to be a Russian steel designation. An international code should be added.
We corrected it.
-line 53: the alloys differ significantly in their behaviour. It should at least be mentioned in this part of the text that Al-Pb is monotectic and shows a miscibility gap also in the melt
We corrected it.
-line 57: (it was important…): the connection of soldering and the processing of large melt batches with EMP seems not really straightforward and should be explained. Electromagnetic pulses and microelectronics go not really well together.
We corrected it.
-line 76: two melts were carried out: revise expression
We corrected it.
-line209: Kirkendall
We corrected it.
Reviewer 3 Report
The standard of English needs to be improved significantly. In some parts, the terms used in the paper are difficult to understand because the wrong words have been used.
The title of the paper is too general and must be changed to reflect the research work reported. The abstract refers to a new method developed by the authors. However, electromagnetic processing of melts is not new and the authors have not described what "new" process has been developed. How does their process differ from other electromagnetic processes? What exactly is the originality? In the introduction, there is reference to increases in density of some of the materials used. It is not clear how density increases. Do the authors want to say that porosity levels decrease or are other changes in microstructure responsible? An explanation is needed. Similar to 3, in the introduction, an increase in mass is mentioned. How is this possible? There is reference in parts of "grinding" of the microstructure. This is not clear. Are the authors suggesting that there is breaking up of particles? It is suggested that there are only two labs in the world working on application of electromagnetic pulsing on melts. This is not so. There are more than that. More details need to be given regarding the generator used. In section 2, "copper forms" are probably copper "moulds." Arrows need to be used in fig.2 to show the different zones. In presenting and discussing the results, I think it will be useful to compare to details from equilibrium phase diagrams. The discussion to suggest that the EMP process is similar to ultrasonic processing in not very convincing. More analysis is needed either way. Why would the two processes be so similar? On its own this is not sufficiently original. I would advise a serious rethink of what the paper is about.Round 2
Reviewer 1 Report
Overall comments did by Reviewers were fixed by authors. However, there is one of them that persists, which is related with the theoretical explanations to justify the results. It is just hypothesis.
In the conclusion was introduced: "unlike ultrasonic processing, it does not have a destructive effect on the furnace equipment". The authors should present evidence of these observations.